# The Influence of Elevation, Land Cover and Vegetation Index on LST Increase in Taiwan from 2000 to 2021

Sahidan Abdulmana [1,*], Matias Garcia-Constantino [2] and Apiradee Lim [3]

1 Department of Information Technology, Faculty of Science and Technology, Fatoni University, Pattani 94000, Thailand

2 School of Computing, Ulster University, Belfast BT15 1AP, UK

3 Department of Mathematics and Computer Science, Faculty of Science and Technology, Prince of Songkla University, Pattani 94000, Thailand

* Correspondence: yee_laal@hotmail.com; Tel.: +66-(0)86-290-5414

**Abstract:** Land Surface Temperature (LST) is an important factor in ground surface energy balance and in universal climatology studies. Elevation, Land Cover (LC), and vegetation index are three factors that influence ground surface variation, and their influences vary depending on geography. This study aimed to: (i) investigate the seasonal patterns and trends of daytime LST, and (ii) examine the influence of elevation, LC, and vegetation index on daytime LST increase in Taiwan from 2000 to 2021. LST, vegetation, and LC data were downloaded from the Moderate Resolution Imaging Spectroradiometer (MODIS) website, and elevation data were downloaded from the United States Geological Survey (USGS) website. The natural cubic spline method was applied to investigate annual seasonal patterns and trends in daytime LST. Linear regression modeling was applied to investigate the influence of elevation, LC, and vegetation index on daytime LST increases. The results showed that the average increase in daytime LST per decade in Taiwan was 0.021 °C. Elevation, LC, and vegetation had significantly affected the daytime LST increase, with $R^2$ of 32.5% and 28.1% for the North and South parts of the country, respectively. The daytime LST increase in the North at elevations higher than 1000 m had an increasing trend, while in the South the increasing trend was found at elevations higher than 350 m above sea level. All types of forest and urban areas in the North had a higher daytime LST increase than the average, while in the South, the areas with water, closed shrubland, and urban parts had a higher daytime LST increase than the average.

**Keywords:** cubic spline function; elevation; land cover; vegetation index; land surface temperature; linear regression

## 1. Introduction

Land Surface Temperature (LST) is the ground temperature of the Earth's surface, resulting from land surface–atmosphere interactions and the fluxes of energy between the surface and the atmosphere [1]. One of the significant benefits of LST is that it can be used as a tool to monitor global and local climate change in order to understand environmental situations and their effect on the sustainability of human life. However, the variation of LST depends on many aspects of the land and its use, such as elevation [2], Normalized Difference Vegetation Index (NDVI) [3], and Land Cover (LC) [4]. Elevation is the height measurement above or below sea level, and it is a fixed geographic reference point used when referring to ground surface points. It is one of the factors that affect climate change. Most of the world's population lives in areas at elevations of 150 m or below [5]. Previous studies have observed the direct effect of elevation on LST from Moderate Resolution Imaging Spectroradiometer (MODIS) data and reported that there was a significant impact of elevation on surface temperature over large regions, especially in low altitude [2].

Another important factor that affects LST is the vegetation index, which is a greenness index of plant health on the ground surface that can be applied to investigate the denseness

of the plants on the ground surface. Furthermore, the vegetation index is critical in the monitoring of energy balances on both the local and global scales [6]. It is used to absorb carbon dioxide and release oxygen to humans and the ecosystem during the day. The variation of vegetation does not only disturb the water cycle and carbon dioxide, but it is also impacting the balance of energy on the Earth's surface and influencing the alteration of surface temperature [7]. Therefore, LC is another factor influencing LST changes [8]. LC is spatial information about the surface cover on the ground for forests, urban areas, wetland, grassland, shrubland, savannas, and water. Diverse color in different types of areas, in satellite images, could be used to point to LC change. The effects of human activities on natural ecosystems, from local to global, will impact surface temperature increase [4].

Previous research works have used a variety of statistical modeling techniques to investigate the factors influencing LST increases. Linear regression analysis was applied in many studies. Patil et al. [9] applied a linear regression model to identify the relationship between LST and NDVI, and to study the impact of Land Use and Land Cover (LULC) changes over LST. A multiple linear regression model was utilized by Prasetya et al. [10] to investigate the impact of the vegetation index, elevation, and LC in Central Sumatra, Indonesia. Prasetya et al. [10] reported that the alteration of LST according to different vegetation indexes is clearly expected. A supervised classification and mono-window algorithm were applied to observe the influence of LC types on LST [11]. Furthermore, Pearson's correlation and linear regression models were used to measure the relationship between LST and LC types [12].

However, Taiwan is a mountainous island in the western Pacific, located in the Tropic of Cancer, where at periods of the year and day the sun is directly overhead. Furthermore, the heat in summer affects people's health (heat stress, symptoms of fatigue, illness, dengue fever, and enteroviral infection) in Taiwan [13]. There are many environmental hazards that are widespread in the urban areas. Vehicle pollution contributes to the smog that may plague large and small cities, such as Taipei, and worsens air conditions in urban and rural areas. Environmental degradation is mainly caused by Taiwan's increase in economy and industrialization. The climate in Taiwan is influenced by two types of monsoons: the summer monsoon from the southwestern direction and the winter monsoon from the northeastern direction [14]. The northern part of Taiwan includes a variety of industries such as petrochemical plants, while the southern region is mainly agricultural, in which vegetables, fruit, and rice are grown. These two areas have the largest population of the country, and there are many growing construction projects, including commercial centers, underground transportation systems, new buildings, and apartments [13]. Therefore, the alteration of elevation, vegetation, and LC might affect daytime LST in Taiwan. There have been a few studies focusing on those factors to observe an association with daytime LST. To investigate these effects, one of the most accessible and reliable sources of LST data is from satellite images. Thus, the objectives of this study are: (i) to examine the seasonal patterns and trends of daytime LST in Taiwan based on MODIS data, and (ii) to analyze the influence of elevation, LC, and vegetation index on daytime LST increase between the North and South of Taiwan.

## 2. Materials and Methods

### 2.1. Study Area

Taiwan is a small island, approximately 394 km long and 144 km wide. The total land area is 35,883 km$^2$, and the highest peak is 3952 m. It is covered by mountains and foothills for two-thirds of the island. It is the fourth-highest island in the world. The average daily temperature range is between 17 and 24 °C (Degrees Celsius). The average temperature in the North is around 21.7 °C, and in the South it is approximately of 24.1 °C. The lowest temperature between January and March is around 10 °C, especially in the mountainous regions. Occasionally, there have been unusual situations that occurred in some years, for instance, snow and frost [15].

The study area, which occupies the entire Taiwan area, was divided into eight super-regions as shown in Figure 1. The area of each super-region is 105 km × 105 km, and they were separated into two parts, each of these consisting of four super-regions: (i) the northern region, consisting of super-regions 1 to 4, and (ii) the southern region, consisting of super-regions 5 to 8.

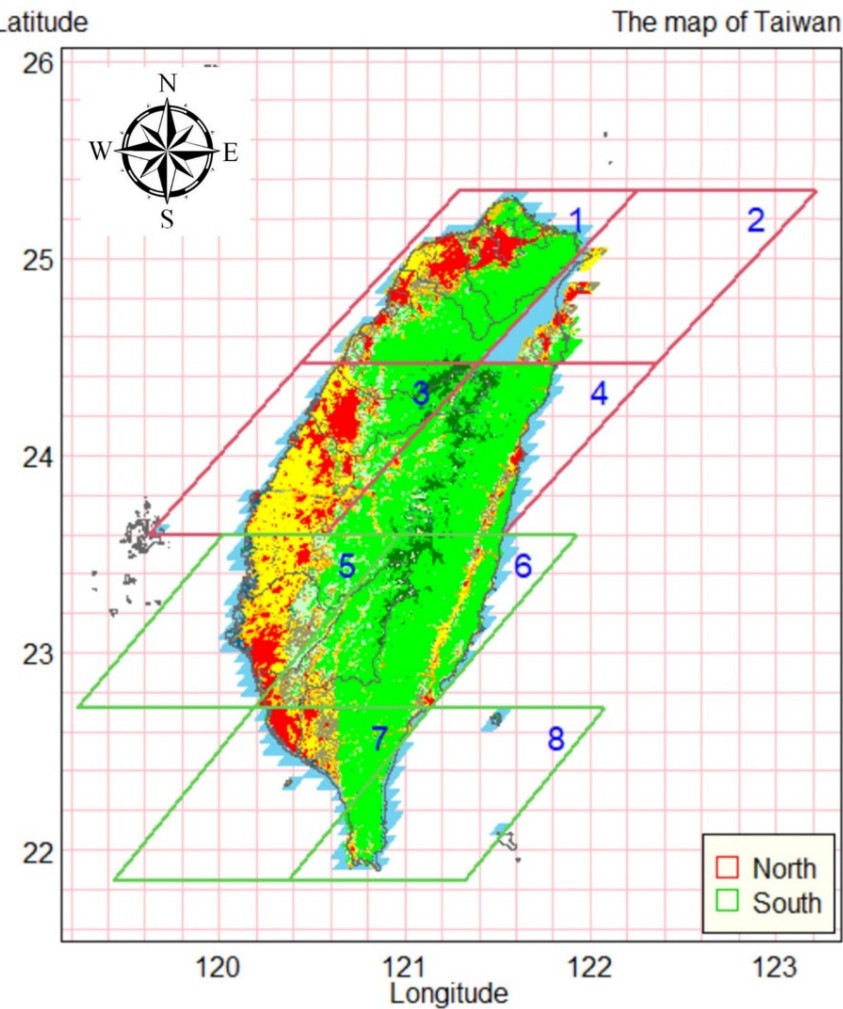

**Figure 1.** The map of Taiwan's super-regions.

Each super-region consists of 25 regions, each one of them with an area of 21 × 21 km$^2$, resulting in nine sub-regions. Each sub-region consists of 7 × 7 km$^2$, or 49 pixels, based on a 1 × 1 km$^2$ grid. Therefore, there were 88,200 pixels in the whole study area. The temperatures were aggregated into pixel levels and used for further analysis.

### 2.2. Data Use

Daytime MODIS/Terra LST and emissivity LST data were downloaded from the MODIS emissivity eight-day composite global product for every km$^2$ for the period (from 18 February 2000 to 31 December 2021). In each region, there are approximately 46 eight-day observations per year, with a total of 966 observations over the last 21 years (46 eight-day observations × 21 years). The name of the product used is Land Surface Temperature and Emissivity (Terra) dataset, 8-Day, 1000 m (MOD11A2). This dataset has better quality, and it provides data on the accuracy of the retrievals [16]. LC data were obtained from MODIS data maps, and these raster maps were transformed to vectors using the QGIS software version 2.18.15. The LC was classified into 18 categories in smaller pixels. The

classifications used for the LC data and their respective percentages in the study area are listed in Table 1.

**Table 1.** LC classification defined by the International Geosphere-Biosphere Programme (IGBP) and the percentage of each area.

| No. | Land Cover | % | No. | Land Cover | % |
|-----|------------|---|-----|------------|---|
| 0 | Water | 5.54 | 9 | Savanna | 7.55 |
| 1 | Evergreen Needleleaf Forest | 3.04 | 10 | Grassland | 1.76 |
| 2 | Evergreen Broadleaf Forest | 48.08 | 11 | Permanent Wetland | 0.99 |
| 3 | Deciduous Needleleaf Forest | 0.00 | 12 | Cropland | 15.37 |
| 4 | Deciduous Broadleaf Forest | 0.05 | 13 | Urban and Built-Up | 6.92 |
| 5 | Mixed Forest | 0.48 | 14 | Natural Vegetation Mosaic | 3.37 |
| 6 | Closed Shrubland | 0.02 | 15 | Snow and Ice | 0.00 |
| 7 | Open Shrubland | 0.00 | 16 | Barren/Sparsely Vegetated Land | 0.32 |
| 8 | Woody Savanna | 6.54 | 17 | Tundra | 0.00 |

Elevation data were downloaded from the United States Geological Survey (USGS) Earth Explorer website [17]. The dataset used in this analysis was the Global Multi-resolution Terrain Elevation Data 2010 (GMTED2010), which represents the latest elevation data and delivers a highly detailed level of global topographic data. The data were grouped into nine levels (shown in Table 2) defined as (i) sea level 0–49 m, (ii) 50–99 m, (iii) 100–179 m, (iv) 180–349 m, (v) 350–599 m, (vi) 600–999 m, (vii) 1000–1499 m, (viii) 1500–1999 m, and (ix) 2000 m and above.

**Table 2.** Elevation levels and the percentages of each group.

| Group | Levels | % |
|-------|--------|---|
| 1 | 0–49 | 30.06 |
| 2 | 50–99 | 4.59 |
| 3 | 100–179 | 4.91 |
| 4 | 180–349 | 7.43 |
| 5 | 350–599 | 9.57 |
| 6 | 600–999 | 12.60 |
| 7 | 1000–1499 | 11.53 |
| 8 | 1500–1999 | 8.512 |
| 9 | 2000 and above | 10.82 |

The NDVI dataset was downloaded from the MODIS website. The name of the data set product is Vegetation Indices (NDVI/EVI) (terra, 16 days, 250 m) (MOD13Q1). The vegetation index data were retrieved from March 2000 to December 2021. The data were recorded every 16 days. In each year, there were around 23 observations per grid cell. In total, 483 observations were made over the course of 21 years. The NDVI data defines values from −1 to 1, as shown in Table 3. The values of NDVI from 0.6 to 1 specify a forest, and 0.2 to 0.5 indicate shrubs and grassland, whereas the values between 0 and 0.1 indicate barren land, snow, rock, concrete, and soil. The negative values specify water areas [18].

**Table 3.** Vegetation index and the percentages of each group.

| Group | Land Types | Levels | % |
|-------|-----------|--------|---|
| 1 | Water | −1––0.1 | 0.1 |
| 2 | Barren land, snow, rock, concrete, and soil | 0.0–0.1 | 17.8 |
| 3 | Shrubs and grassland | 0.2–0.5 | 12.3 |
| 4 | Forest | 0.6–1 | 69.8 |

*2.3. Statistical Methods*

A cubic spline model is a spline function constructed from a piecewise third-order polynomial where the second derivatives of each polynomial are normally set to zero at the endpoint. It provides the smoothest model between all the functions and is usually applied to smooth different types of data in study areas, for example, satellite-based time series data [19], interactive computer graphics [20], and real-time digital signal processing [21]. The daytime LST data from 2000 to 2021 were analyzed separately for the northern and southern regions.

In this study, the cubic spline method was applied to extract the annual seasonal patterns of daytime LST for all sub-regions. From the model fitting, a cubic spline model with eight knots with a high adjusted $R^2$ and low cross-validated error was chosen. To obtain the seasonal patterns, the knots were fixed at Julian days 10, 40, 80, 130, 250, 310, 345, and 360. To our best knowledge, the seasonal pattern is studied by studying the whole months' data for a particular season in order to ensure the data are stable and show continuous seasonal patterns between years. The daytime LST was adjusted for seasonal patterns using least-squares regression to fit natural cubic spline functions with a further boundary condition ensuring that the slope of the seasonal pattern was the same at the beginning and at the end of each year. Seasonally adjusted daytime LST was calculated by subtracting the original values from the fitted values from the natural cubic spline and adding the mean values of each sub-region back to confirm that the average temperatures over the past 21 years were not changed.

LST plots by day of the year in each sub-region were created to illustrate the seasonal patterns. The seasonally adjusted LST by year in each sub-region was plotted to show the trends. The LST trends of all the sub-regions were plotted on a map of Taiwan to illustrate the different temperature trends for the whole country. From the plot, the LST trends were categorized into five patterns: (i) accelerating increase, (ii) decelerating increase, (iii) stable, (iv) accelerating decrease, and (v) decelerating decrease. For each sub-region, the average increase in daytime LST (°C per decade) and the accelerating increase in LST (°C per decade) were calculated. All the average increases in daytime LST from the 1800 sub-regions were plotted against the accelerating increase in LST per decade. Finally, linear regression was applied to predict the average increase of daytime LST based on elevation, vegetation, and LC. The coefficients from the model were converted to average daily increases of LST with 95% confidence intervals. Confidence interval graphs were created to illustrate the influence of elevation, LC, and the vegetation index on the increase in daytime LST. All statistical and graphical analyses were performed using the R programming language.

## 3. Results

*3.1. Seasonally Adjusted Daytime LST*

Figure 2 shows the seasonally adjusted daytime LST and the values of the increase per decade (Inc/dec) denoting increases and decreases in temperature (°C) per decade from 2000 to 2021 based on the models constructed for this study. The curves in the lowest right-hand panel show the fitted models for all nine sub-regions, indicating the different trends in each sub-region. The Northwest sub-region had the highest temperature of around 32 °C. The temperature trends in almost all sub-regions had slightly increasing trends from 2015 to 2021, except in the Northwest sub-region, which showed a stable trend.

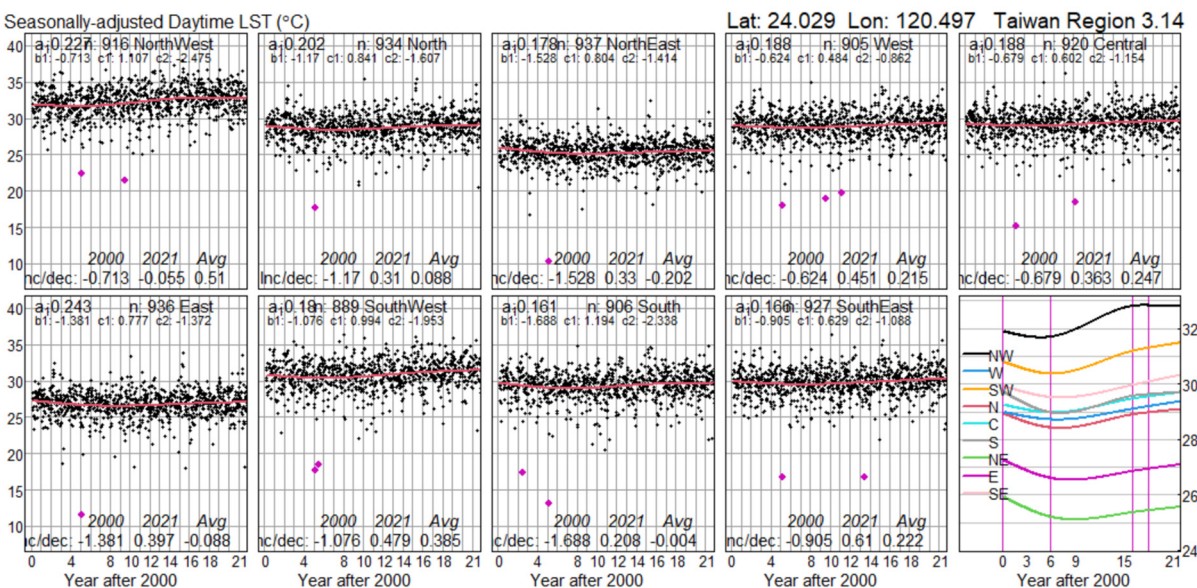

**Figure 2.** Time series data with fitted quadratic polynomials for seasonally adjusted daytime LST in Region 14 of Super-region 3.

### 3.2. Daytime LST Trends for 21 Years

After obtaining seasonally adjusted data for each sub-region, the data were fitted to a second derivative polynomial model to display the daytime LST quadratic trends for 21 years. The quadratic curves of temperature in each area were categorized into five groups based on their shapes, as shown in Figure 3, which illustrates the trends of daytime LST over the 21 years studied in Taiwan separated into the North and South parts of the island.

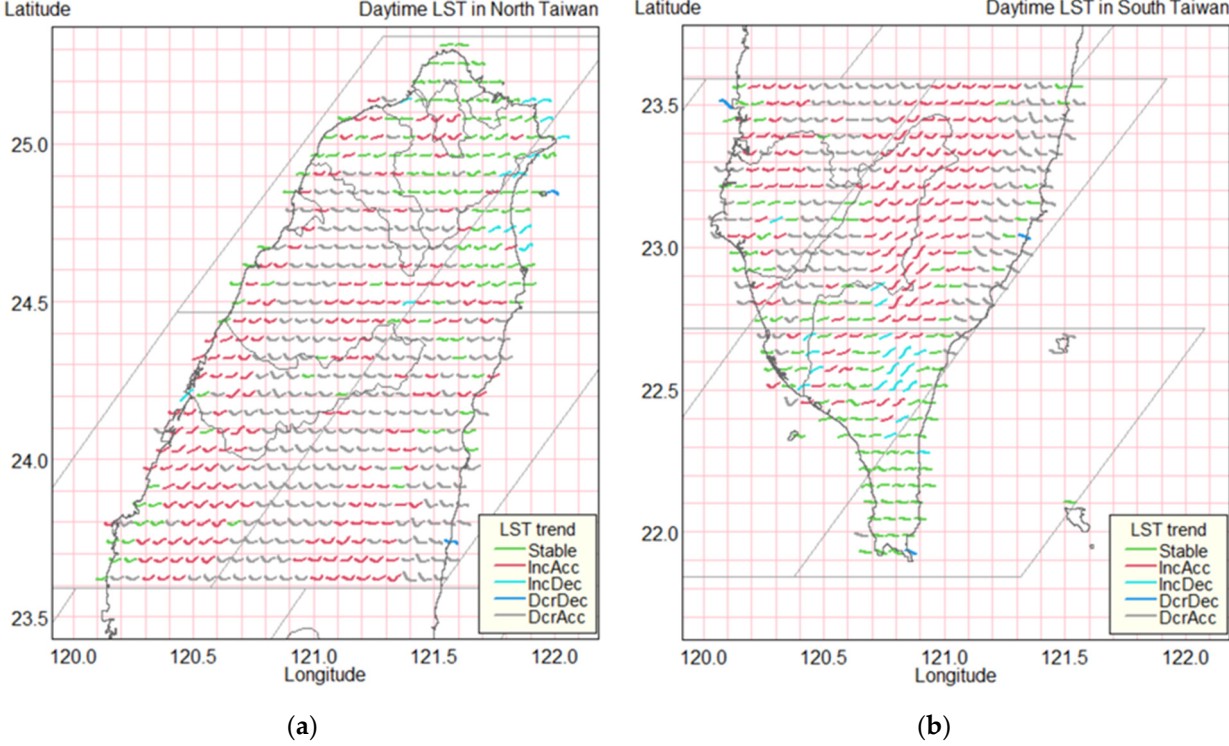

(**a**)            (**b**)

**Figure 3.** Daytime LST trends in Taiwan: (**a**) in the North; (**b**) in the South. Note: IncAcc is accelerating increase, IncDec is decelerating increase, DcrDec is decelerating decrease, and DcrAcc is accelerating decrease.

Figure 3 shows the trends in daytime LST estimated from the linear models between the North and South parts of Taiwan, categorized into five trends represented by different colors: (i) green represents a stable trend, (ii) red an accelerating increased trend, (iii) light blue a decelerating increased trend, (iv) dark blue a decelerating decreased trend, and (v) pink an accelerating decreased trend. The plots were further categorized into binomial terms, acceleration (accelerating increase or decelerating increase) and nonacceleration curves based on the overall change of LST, i.e., whether it was positive or negative. Figure 4 below shows a scatter plot of the average accelerating increase of daytime LST ($°C/decade^2$) and the average daytime LST increase ($°C/decade$) in each sub-region.

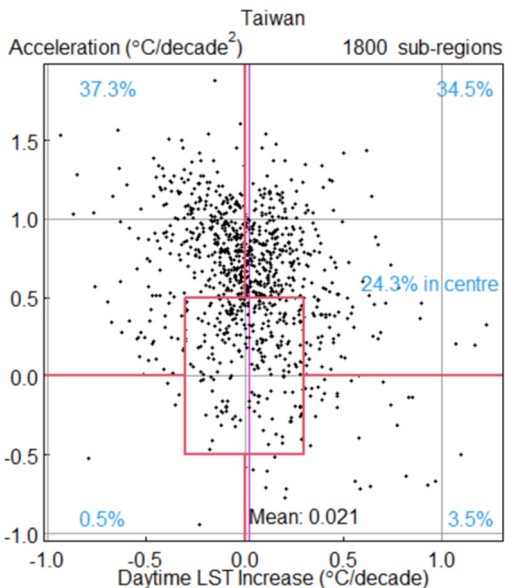

**Figure 4.** Scatter plot of the estimated acceleration and deceleration in increase in daytime LST and increase in daytime LST in Taiwan over the 21-year period from February 2000 to December 2021.

Figure 4 presents the estimated acceleration and deceleration in the increase of daytime LST in Taiwan over the past 21 years and summarizes the trends per decade. The vertical, horizontal, and square lines in red delineate separate categories of temperature change from zero defined by the combination of temperature (increase or decrease on the X-axis, and acceleration and deceleration on the Y-axis) from which the choropleth map above, in Figure 3, was created. The trends in surface temperature in Taiwan have had an accelerating decrease trend (37.3%) for all sub-regions, followed by an accelerating increase of 34.5%, a stable trend (24.3%), a decelerating decrease trend (0.5%), and a decelerating increase trend (3.5%). The pink vertical line shows an overall temperature increase of 0.021 °C per decade.

### 3.3. The Average NDVI Increase per Decade

Figure 5 illustrates the average and initial increase of the vegetation index per decade between the North (a) and South (b) of Taiwan. The X-axis represents the average of the vegetation index, and the Y-axis shows the NDVI initial increase per decade. The top right panel shows the mean NDVI increase per decade with a percentage, and the bottom left panel represents the NDVI increase per decade. The dark green color represents the highest NDVI increase per decade, followed by the decrease in the light color. The vertical and horizontal red lines delineate separate categories of mean NDVI increase per decade. The plot shows that the NDVI in the northern region had a stable vegetation index per decade, which was 57.9%. The South region shows the mean NDVI increase per decade for all types of mean increase. The highest mean vegetation increase is 0.19 per decade, which shows 34.4%.

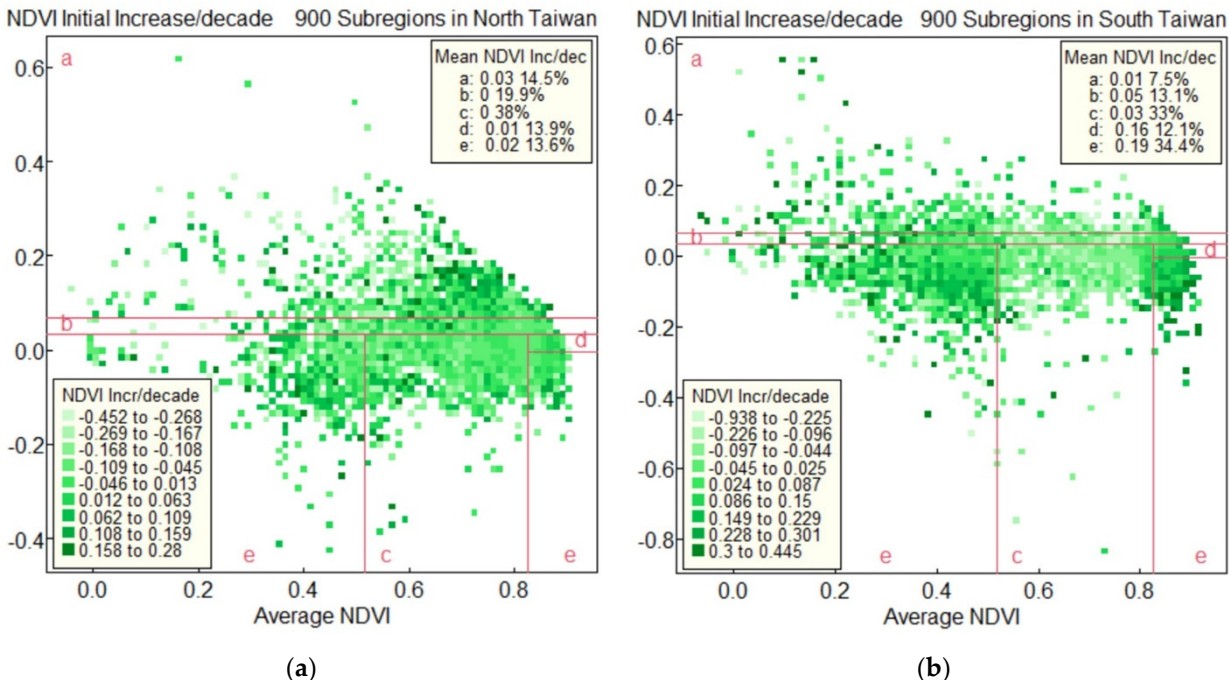

**Figure 5.** A plot of average and initial increase of NDVI per decade: (**a**) the North; (**b**) the South of Taiwan.

### 3.4. The Influence of Elevation, LC, and Vegetation Index on Daytime LST Increase

Figure 6 illustrates the daytime LST increase per decade by elevation, LC, and vegetation index in the North (a) and in the South (b) regions. The Y-axis represents the average daytime LST increase per decade at temperature (°C), and the X-axis shows elevation, LC, and vegetation index as predictors. The horizontal lines represent the overall daytime LST increase. The percentages of $R^2$ were 32.5 in the North and 28.1 in the South. The percentages of $R^2$ from the crude analysis in the North were 29.1 for elevation, 21.8 for LC, and 0.4 for vegetation, while in the South they were 25.5 for elevation, 14.9 for LC, and 6.5 for vegetation. Elevation, LC, and vegetation were significantly associated with daytime LST increases in both the North and South regions.

In the North region, daytime LST increase at elevations less than 100 m above sea level was significantly higher than the average (having to increase LST), while daytime LST at elevations of 180–1499 m was significantly lower than the average (having to decrease LST). A decreasing trend of daytime LST increase was found at elevations less than 1500 m, while an increasing trend of LST increase appeared at elevations equal to or greater than 1500 m above sea level. Water, evergreen needleleaf forest, deciduous broadleaf forest, permanent wetland, cropland, and urban areas had significantly higher daytime LST increases than the average for the LC, whereas evergreen broadleaf forest, woody savannas, savannas, a natural vegetation mosaic, and barren or sparsely vegetated land had significantly lower daytime LST increases than the average. For vegetation, there was a fluctuating NDVI pattern across daytime LST increases.

In the South region, daytime LST increase at elevations less than 600 m above sea level was significantly lower than average, while LST increase at elevations greater than 600 m was significantly higher than average. At elevations of 350 m or higher, there was an increasing trend of daytime LST. For the LC, areas with mixed forest, closed shrubland, cropland, and urban parts had significantly higher daytime LST increases than the average, while the other types of LC had significantly lower daytime LST increases than the mean. For the vegetation index, the increasing trend of daytime LST increase was found at high vegetation patterns, which were higher than the average of daytime LST increase.

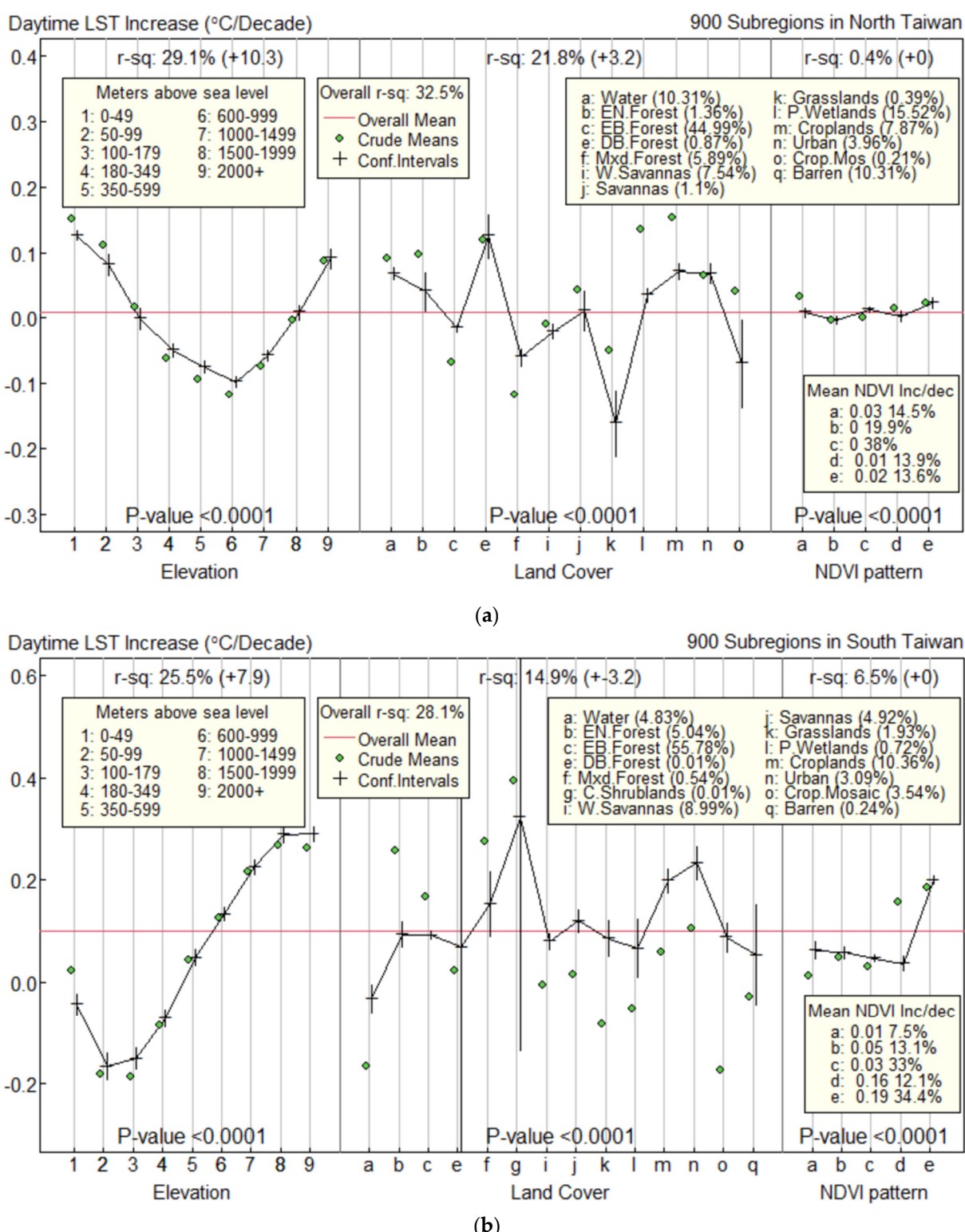

**Figure 6.** Confidence intervals for daytime LST increased in all sub-regions—(**a**) in the North regions, and (**b**) in the South regions of the study area—based on a linear regression model with elevation, LC, and vegetation index as determinants.

## 4. Discussion

In this study, the cubic spline method with an applicable number of knots was successfully able to demonstrate the annual seasonal daytime LST patterns and trends. The results showed that the trends in daytime LST have an accelerating increase of 34.5% in all

sub-regions. The overall mean daytime LST increased by 0.021 °C per decade. This finding is parallel with the previous analysis by Abdulmana et al. [22]; Abdulmana et al. [23], that Taiwan's daytime LST had increased. Moreover, our findings are consistent with the Council for Economic Planning and Development in Taiwan report [24], which reported that Taiwan's temperature in the years 1980–2009 had increased by 0.29 °C per decade. LST increase in our study is lower than the global mean temperature increases with a rate of 0.07 °C per decade. This could be because more human activities have caused the temperature to rise in the last four decades than it did two decades later. Furthermore, the average daytime LST increase in the North was 0.01 °C per decade, which indicated an increase in LST in this area, while the average daytime LST increase in the South was 0.10 °C per decade, which indicated an increase in LST in this area. This finding was consistent with Hsu and Chen [25], who discovered that temperature in the North of Taiwan tended to rise while LST in the South tended to fall over the past century. This difference might be due to the implementation of government policies aimed at reducing greenhouse gases and global warming [26].

In our study, elevation was significantly associated with a daytime LST increase. In the North, daytime LST increase at elevations less than 100 m above sea level was significantly higher than the average, while in the South, daytime LST increase at elevations equal to or greater than 600 m was significantly higher than the average. Furthermore, LST increase in the North had a decreasing rate at elevations greater than 180 m and started to have an increasing rate at elevations equal to or greater than 1500 m above sea level, whereas LST increase in the South started to have an increasing rate at elevations of 350 m and higher. Many previous studies had reported that LST increased with the increase of elevation [2,27,28]. This is because the rise in LST is due to the transformation of forest and mountains as a result of deforestation on the mountains in order to conduct agriculture, for example by growing fruit, rice, and vegetables, particularly in the southern region, which is mainly an agricultural area [15]. This deforestation had an effect on the daytime LST increase at high altitudes. This is one of the reasons why the daytime LST in the South had a higher daytime LST increase than in the North.

The daytime LST increase depends on the different types of LC, and a rising LST was found in urban areas in both regions. Our study was consistent with previous analysis by Ayanlade et al. [29], which reported that LST increase in urban and built-up areas [30]. In Taipei, which is the capital city of Taiwan, the daytime LST increases with the increase of the urbanization index [31]. In the North, water, evergreen needleleaf forest, deciduous broadleaf forest, permanent wetland, cropland, and urban areas had significantly higher daytime LST increases than the average, while the area with evergreen broadleaf forest, woody savannas, savannas, a natural vegetation mosaic, and barren or sparsely vegetated land had significantly lower daytime LST increases than the average. In the South, areas with mixed forest, closed shrubland, cropland, and urban parts had significantly higher daytime LST increases than the average, while the other types of LC had significantly lower daytime LST increases than the average. Therefore, in the South of Taiwan, LST increased in high altitude, mixed forest, closed shrubland, and in areas of high mean NDVI increase because of the history of Typhoon Morakot, which destroyed much of Taiwan in 2009, especially affecting the South [32]. The typhoon was the cause of the transformation of the hillside and base of the valley covered by dense vegetation, which became the rocky and bare soil surface that caused the increase of its ground temperature in this part of Taiwan.

However, these results are in agreement with several previous studies that found that LC change can cause an increase in LST [8]. Changes in LC cause changes in both local and global climate, and they have a significant impact on LST [33]. Furthermore, this finding supports the results of the study conducted by Karakuş [34], which specified that the LST variation depends on the LC types, and that LST generally rises in built-up urban areas while experiencing a reducing trend in rural areas. However, the reason for the increase in LST in different LC types between the North and the South has to be investigated for further study.

## 5. Conclusions

Results presented in this study prove that the use of natural cubic spline functions to explore the seasonal patterns and trends of daytime LST and multiple linear regressions to investigate the influence of elevation, LC, and vegetation index on daytime LST increase is possible. Daytime LST increase in Taiwan has shown an increasing trend in the past two decades. The overall surface temperature has increased by 0.021 °C per decade. The trends in surface temperature in Taiwan have shown an accelerating increase of 34.5%, which confirms the surface temperature in Taiwan has increased. Elevation, LC, and vegetation index all had different effects on daytime LST increase between the North and South. In the North region, LST increases at low elevations, while in the South region, it increases at high elevations. The influence of elevation, LC, and vegetation index varied depending on the diverse geography. The urban area showed an increasing surface temperature in both regions, indicating that urbanization's expansion affects LST increases. The increase in the vegetation index in forest areas and shrubland in the South part showed an increase in LST trends, indicating that LST is increasing at high elevations. This is because in the southern region there is a mainly agricultural area where rice, fruit, and vegetables are grown. Deforestation to replace the vegetation is causing the surface temperature to rise. The results of the study could be provided for the Taiwan government to warn Taiwan's policymakers to better plan for and deal with future global warming. Therefore, other factors influencing the increase in daytime LST apart from elevation, LC, and vegetation index, for instance, rainfall and land use (LU), should be investigated in further studies to monitor anthropic climate change and effectively control and manage land use and its effect on LST.

**Author Contributions:** Conceptualization, S.A. and A.L.; methodology, S.A. and A.L.; investigation, S.A.; formal analysis, S.A., M.G.-C. and A.L.; resources, S.A.; writing original draft, S.A.; writing review and editing, M.G.-C. and A.L. All authors have read and agreed to the published version of the manuscript.

**Funding:** This research received no external funding.

**Institutional Review Board Statement:** Not applicable.

**Informed Consent Statement:** Not applicable.

**Data Availability Statement:** Not applicable.

**Acknowledgments:** We acknowledge the Department of Information Technology, Fatoni University, and the School of Computing, Ulster University, Belfast, United Kingdom for providing facilities for this study. We are grateful to Don McNeil for his immense guidance during the analysis.

**Conflicts of Interest:** The authors declare no conflict of interest.

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
