# Peer review of "The Influence of Elevation, Land Cover and Vegetation Index on LST Increase in Taiwan from 2000 to 2021"

_sustainability, doi:10.3390/su15043262_

Round 1
Reviewer 1 Report
Review on
The influence of elevation, land cover and vegetation index on LST increase in Taiwan from 2000 to 2021
The paper is well written. With a single example (row 49), minor English language review is needed.
Still, besides elevation, LC and vegetation, could other factors influence LST increase?
In the introduction section, row 33, LST should be defined from a basic source, not 2018.
43-44 – This conclusion is a clear classical one, specified in any climatology book. There are even classic indices indicating the relation between rainfall quantities or temperature and elevation. The source cited could be “sent in time”.
Rows 54-55…I think that the overall syntax should be changed. Land cover is a type of physical land? Sattelite images can have diverse color, but usually they are multi or hyperspectral. The value of the pixels is interesting, not the color.
Rows 56-58 … the conclusions can be dated back to 1880 at least and are found in any geography or environmental science graduate book.
Row 65…. same as above, an alteration of LST according to different vegetation indexes is clearly expected.
Despite the depiction of the super-regions (quite interesting term, I would gladly like a response from the authors how would they define super-regions, for example inside USA, Russia or Ukraine at least). Also, on what was based the criterion of dimension in separating super-regions, regions?
Row 92 … ºC is the same as “Celsius degrees”.
Reference source 16, row 384-385 should be reconsidered. I am sure a book describing the geography of Taiwan can be found.
For references such as row 391, better refer to Tarpley et al. and Kogan, or Huete, K. Didan, T. Miura, E. P. Rodriguez, X. Gao, L. G. Ferreira.
Author Response
Dear reviewer,The attached files are responses to comments and the manuscript marked up using the “Track Changes”.
Best regards,
Sahidan

Reviewer 2 Report
Dear Authors,
I strongly believe that this paper has potential!
However, there are a lot of issues that require addressing. Entire paper needs improvement. I don't mean about 'English language'. Sentences used in this paper are in general not as 'scientific language in papers' should be.
1. 'Furthermore' is used really often!
2. Introduction - line 48: 'vegetation index is critical for...' - index is number value that shows something. It can not be 'critical for energy balances'. Its value can be 'critical' in 'monitoring' of energy balances, or 'assessment of'...
Line 49 - again 'Furthermore'.
Line 49 - 'it is used to absorb carbon dioxide...' - from this sentence one can conclude that 'vegetation index is used to absorb carbon dioxide...'!? What did you wanted to point to?
Line 55: 'Satellite images with ...' Sentence should be rephrased: 'Diverse color in different types of area, in satellite images, could be used to point to LC change'
Line 55: 'Furthermore, the effects of human activities on natural ecosystems, from local to global, are clearly influenced by the environment'
'Furthermore' again.
The effects of human activities are influenced 'BY' the environment!? Also this sentence is does not present continue of the 'thought' written in previous one.
Line 59: 'Previous research'. Whose previous research? Research conducted by same authors that wrote this paper? Or by other scientific community. It should be clearly stated and cited.
Line 65: 'They' reported... Who are they? This should not be the proper way to address statement in scientific paper.
Line 70: '... certain times of the year and day...' 'Periods' of the year and day is proper what to say.
Line 72: 'The highest peak...'
Is the highest peak in central ridge something that Taiwan climate is influenced by? Why its hight is important in correlation with monsoons? This way the sentence is not clear at all.
Line 75: 'In the northern part of Taiwan...'
It should be something like: 'The northern part of Taiwan includes variety of industries such as.. while southern region is...'
Line 83: '... reliable resources...' Is this personal opinion or a fact?
2.1. Study area:
In Introduction there is a lot of geographical properties of Taiwan. They are repeated here also! Maybe it would be better for Study area to include geog. facts and Introduction to include only facts that are related to this research (such as living conditions... the ones that this study is focused to.
Line 91: and again 'Furthermore'...
Line 97: "The study area, which consisted of the whole of Taiwan" - 'which occupies entire Taiwan area' is better.
In general, there are a lot of conesecutive sentences which are not continuation of though of previous ones.
Conlcusion - line 324: 'The use of a natural cubic spline function to explore the seasonal patterns and trends of daytime LST and multiple linear regression to investigate the influence...'
Better to write something like: 'Results presented in this study proves that use of natural cubic spline function to explore .... is possible.'
This are just some of the things (there are more) which I've wanted to point to, so that you can understand what should be done to make this paper much better.
I would recommend for major revision of this paper because I believe that is has good potential. I hope that you will make it much, much, better with clearer thoughts in each sentence.
Regards
Author Response
Dear reviewer,The attached file is a response to comments.
Best regards,
Sahidan

Reviewer 3 Report
Thank you very much for submitting your manuscript to Sustainability. Manuscript requires a major revision to make is publishable. Here are my specific comments:
1. Please replace r squared by R2
2. Please define the novelty and the utility of the work.
3. Please describe the discussion of the results in detail.
4. What are the future scopes of the study?
Thanks
Author Response

(The authors gave the same response as above.)

Round 2
Reviewer 2 Report
Dear Authors,
Thank you for making this paper even better!
As far as I am concerned, this paper is ready to be published.
Keep up good work.
Regards
Reviewer 3 Report
The paper is ready for publication after significant revision.